# Astrocyte–Neuron Interaction via the Glutamate–Glutamine Cycle and Its Dysfunction in Tau-Dependent Neurodegeneration

**DOI:** 10.3390/ijms25053050

**Published:** 2024-03-06

**Authors:** Marta Sidoryk-Węgrzynowicz, Kamil Adamiak, Lidia Strużyńska

**Affiliations:** Laboratory of Pathoneurochemistry, Department of Neurochemistry, Mossakowski Medical Research Institute, Polish Academy of Sciences, 5 Pawińskiego Str., 02-106 Warsaw, Poland; kadamiak@imdik.pan.pl (K.A.); lidkas@imdik.pan.pl (L.S.)

**Keywords:** astrocyte–neuron integrity, glutamate–glutamine cycle, tau-dependent neurodegeneration, glutamine transporters, glutamate transporters

## Abstract

Astroglia constitute the largest group of glial cells and are involved in numerous actions that are critical to neuronal development and functioning, such as maintaining the blood–brain barrier, forming synapses, supporting neurons with nutrients and trophic factors, and protecting them from injury. These properties are deeply affected in the course of many neurodegenerative diseases, including tauopathies, often before the onset of the disease. In this respect, the transfer of essential amino acids such as glutamate and glutamine between neurons and astrocytes in the glutamate–glutamine cycle (GGC) is one example. In this review, we focus on the GGC and the disruption of this cycle in tau-dependent neurodegeneration. A profound understanding of the complex functions of the GGC and, in the broader context, searching for dysfunctions in communication pathways between astrocytes and neurons via GGC in health and disease, is of critical significance for the development of novel mechanism-based therapies for neurodegenerative disorders.

## 1. Astrocyte–Neuron Crosstalk in Normal Brain Function

Astrocytes represent the largest group of glial cells, and in some regions of the human brain, they may predominate in number over neurons [1], encompassing up to 40% of all cells [2]. It is well established that astrocytes are involved in many functions in the central nervous system (CNS), and considerable evidence exists supporting their crucial role in brain homeostasis. For instance, astroglia are involved in brain plasticity, neurotransmission and neuromodulation, the maintenance of the blood–brain barrier (BBB), the stabilization of cerebral blood flow, and extracellular ion balance. They also participate in glial scar formation during brain injury. Crosstalk between astrocytes and neurons is essential during brain development in many processes such as synaptogenesis, synapse elimination, and structural plasticity through a variety of pathways [3,4,5]. 

Through several identified pathways (e.g., exocytosis, diffusion through transmembrane pores, or active transport across the membrane via specific carriers), astrocyte secrete a wide range of factors such as chemokines, cytokines, neurohormones, neurotransmitters, and small metabolites such as nucleosides, nucleotides, and amino acids [3]. The metabolism of astrocytes is tightly coupled with the metabolism of neurons [5], and interactions between these cells are critical for maintaining the homeostasis of neurotransmitters such as the most abundant excitatory neurotransmitter, glutamate (Glu), and a major inhibitory neurotransmitter, γ-aminobutyric acid (GABA) [6,7].

### Significance of GGC Cycle 

Considerable evidence supports astrocytes’ involvement in the metabolism and homoeostasis of major neuroactive amino acids and in mediating the vast majority of these neurotransmitters’ signaling. Notably, key enzymes involved in metabolic reactions related to the turnover of the neurotransmitters Glu and GABA are highly active in glia. In neurons, glutamate is relocated and stored in synaptic vesicles through specific vesicular glutamate transporters [8]. Once glutamate is packaged into the synaptic vesicles and stored in the synaptic bouton, it is ready to be released upon the arrival of an action potential, which will induce the opening of voltage-dependent calcium channels, increasing the intracellular calcium (Ca^2+^) concentration. The Ca^2+^ influx facilitates vesicle fusion with the plasma membrane, which releases the neurotransmitters into the synaptic cleft of the tripartite synapse [9]. The major Glu transporters are located on the astroglia or post-synaptically [10,11]. This is in agreement with the results from studies with the radiolabeled precursor that Glu and GABA are recycled through Gln metabolism in astroglia [12] and is further supported by findings regarding Gln metabolizing enzymes, such as glutamine synthetase (GS), a specific enzyme that catalyzes its synthesis in glia [13] and phosphate-activated glutaminase (PAG), which deaminates Gln to Glu in nerve endings [14]. Glu released from the glutamatergic synapse is taken up into astrocytes by glutamate transporters and metabolized to Gln by GS [15]. Gln is then released into the extracellular space, and after secretion, it is taken up into neurons and converted back to Glu by PAG [16]. GABA released from the GABAergic synapse is translocated to astroglia and catabolized to the tricarboxylic acid cycle (TCA) intermediates. Gln is also metabolized via the TCA from succinate and the subsequent synthesis of α-ketoglutarate and glutamate. Glu formed by PAG activity in the GABAergic neurons is converted by glutamate decarboxylase (GAD) to GABA (Figure 1).

It is worth noting that GS is expressed exclusively in astrocytes [15], while GAD is present only in GABAergic neurons and not in astrocytes [17]. In addition, it is of functional importance that the activity of PAG is higher in neurons than in astrocytes [18,19,20]. As mentioned, the glutamate–glutamine cycle (GGC) mediates the clearance of Glu from the synaptic cleft by uptake into astrocytes and the subsequent amidation of Glu into Gln, which is then transferred into neurons for the re-synthesis of Glu. Hence, the cycle leads to a net transfer of nitrogen from the astrocytic to the neuronal compartment. In order to maintain nitrogen homeostasis in the microenvironment, i.e., the pre- and post-synaptic neurons and the surrounding astrocyte, this nitrogen is transferred back to the astrocyte.

There is considerable evidence supporting the role of GGC in neurodegenerative disorders. The use of short-chain fatty acids (SCFAs) on a trangenic mice model allows us to better understand the processes relevant to the dysfunction of GGC appearing in neurodegeneration [21]. The SCFAs (mainly acetate, propionate, and butyrate) produced by the bacterial fermentation of dietary fiber were recognized as the pivotal molecules regulating several neurodegenerative diseases. Long-term dietary SCFA supplementations at the early aging stage of APPswe/PS1dE9 transgenic mice enhanced astrocyte–neuron communication via GGC and alleviated cognitive impairment by reducing Aβ deposition and tau hyperphosphorylation. In animal studies, SCFAs showed a positive effect on the glutamate–glutamine cycle and brain energy metabolism [22].

## 2. GGC-Involved Amino Acid Transporter Systems

### 2.1. Subfamily of Glutamate Transporters

Glutamate transporters display an unusual structural motif of 8TM segments and two re-entrant loops. The activity of theses carriers located on both neurons (predominantly EAAT3, 4, and 5) and astrocytes (predominantly GLAST/EAAT1 and GLT1/EAAT2) serves, depending on their location, to maintain low ambient extracellular concentrations of Glu, thus protecting against excitotoxicity [14]. EAAT4 and EAAT5 carries are less abundant, and the majority of their expression is observed by Purkinje cells in the cerebellar molecular layer and retinal Müller cells, respectively [23]. The inward translocation of Gln is coupled to the co-transport of the Na^+^ and H^+^, with K^+^ being anti-transported [24,25]. This mediates the uptake with a massive driving force capable of maintaining an inside-out concentration gradient [26]. Glutamate transporters regulate excitatory neurotransmission and provide Glu for metabolism through the GGC. The Na^+^/K^+^-ATPase that maintains the ion gradients driving transport has been demonstrated to be associated with the GLT1 and GLAST transporters [27,28]. For the purpose of the present discussion, these carriers are the most important because they appear to be primarily involved in Glu uptake into astrocytes throughout the brain, although with some regional variations [23,29].

### 2.2. Glutamine-Recognizing Systems and Carriers

Transporters are integral membrane proteins characterized by the ability to mediate the movement of small hydrophilic molecules across the membrane against their concentration gradients [30]. This process is usually energized by coupling with one or more ions transferred down their electrochemical gradients [31]. Four principal transporting systems account for Gln uptake by mammalian cells. The A, ASC, and N systems catalyze Na^+^-dependent net uptake or release, whereas the N system mediates Na^+^-independent exchange rather than the net uptake of amino acids. The differences in the substrate preferences of each system allow the study of single system-specific transporter functions in vitro using the system-selective uptake of competitive amino acids together with radiolabeled Gln. For example, 2-methylaminoisobutyric acid (MeAIB), threonine (Thr), and leucine (Leu) might be used as competitive substrates of system N; MeAIB, Leu, and histidine (His) act as system ASC competitors; MeAIB, Thr, and His act as system L competitors; and His, Leu, and Thr act as system A competitors (Table 1).

The substrates preferred by the ASC system are alanine (Ala), serine (Ser), cysteine (Cys), and other small neutral amino acids (NAAs) [36]. The transporter isoforms ASCT1 and ASCT2, belonging to the ASC system, are obligatory exchangers [37,38]. ASCT2, unlike ASCT1, is an efficient mediator of glutamine hetero- and homo-exchange [39]. Moreover, ASCT2 mRNA is abundantly expressed in astrocyte cultures, as well as in developing brains [39]. However, only low levels are detectable in the adult brain, suggesting that the increased expression of ASCT2 may be associated with reactive astrocytosis. It has been demonstrated that the transport activity of the ASC system may be responsible for the release of Gln from glia [39]. However, this system does not appear to be crucial for the Gln shuttle from glia to neurons since the activity of the ASCT 1 carrier is relatively weak in glia [39]. The L system has been shown to be involved in both the efflux and influx of amino acids [36]. The LAT1 carrier representing this system is ubiquitously expressed with broad substrate specificity and a preference for bulky, hydrophobic amino acids, while the LAT2 transporter much more efficiently recognizes zwitterionic amino acids such as glutamine [35]. Although the L system only partially participates in Gln translocation within the CNS via the low-affinity, high-capacity uptake of this amino acid, the expression and function of system L transporters have been observed in both astrocytes and neurons [40,41]. LAT2 was observed to mediate both antiport and uniport exchange [35], whereas LAT1 functions as an obligatory exchanger [42,43].

SNAT1 is predominantly present in neurons, with widespread expression in the brain and spinal cord [44]. The expression of SAT2/ATA2 variants was demonstrated by in situ hybridization in the pyriform cortex, hippocampus, and cerebellar granule layer [45]. At higher magnification, SNAT2 expression was detected in cortical and hippocampal neurons but also in glial cells of the posterior commissure and in the choroid plexus [45]. In primary cerebellar cultures, immunoreactivity was present in neurons but not in astrocytes [46].

The N transporter system is distinguished by narrow substrate specificity, which is limited largely to Gln, Asp, and His, and restricted tissue distribution compared to other transport systems [47]. This system is represented by two main amino acid carriers, namely SNAT3 and SNAT5. SNAT3 expression is more pronounced in the neocortex, cerebellum, olfactory bulb, and brainstem [48], while SNAT5 is most enriched in the striatum, spinal cord, and neocortex [34]. SNAT5 has broader substrate specificity than SNAT3, including amino acids such as Gly, Ser, Ala, and Cys. It has been suggested that SNAT5 may also mediate the leakage of glycine from glial cells, which acts as an important co-agonist of NMDA receptors [34]. SNAT3 protein functions as a Na^+^-glutamine cotransporter coupled to the countertransport of H^+^ [49]. Transport is unaffected by membrane depolarization, indicating electroneutral stoichiometry. The expression of SNAT3 is prominent in astrocytes, specifically in perisynaptic glial processes, but no evidence of SNAT3 expression in neurons was found [49]. SNAT3 allows glia to release Gln, which may then be taken up by various protein(s) located in nerve terminals to resynthesize Glu and GABA [50].

## 3. Tau-Dependent Neurodegeneration

### 3.1. Tau: Its Biological Configurations and Involvement in Neurodegeneration

A common feature of human neurodegenerative diseases is the presence of abundant assemblies with the formation of inclusion fibrils [51,52]. These inclusions consist of a single protein that undergoes a transformation from a soluble to an insoluble filamentous form, and their expression is largely confined to central and peripheral nerve cells enriched in axons [53].

The presence of neurofibrillary tangles (NFTs) composed of the microtubule-associated protein tau is one of the hallmarks of AD, frontotemporal dementia with Parkinsonism linked to chromosome 17 (FTDP17), and several other neurodegenerative diseases, including progressive supranuclear palsy (PSP), globular glial tauopathy (GGT), corticobasal degeneration (CBD), argyrophilic grain disease (AGD), or aging-related tau astrogliopathy (ARTAG), collectively known as tauopathies [5,54]. 

In the adult human brain, the six tau isoforms are expressed as a range from 352 to 441 amino acids and as a consequence of the alternative mRNA splicing of transcript from *MAPT* gene consist of 16 exons. These isoforms are natively unfolded and are represented by the presence or absence of inserts in the amino-terminal half and the inclusion, or not, of the 31 amino acid repeat coding by exon 10 of the carboxy-terminal half. Inclusions of exon 10 give rise to three tau isoforms having four repeats (4R), while further exclusions give rise to three isoforms having three repeats each (3R) [55,56]. In AD and several other tau-dependent disorders, all six tau isoforms are detected in the pathological filaments [57]. In other diseases like PSP, GGT, CBD, AGD, or aging, ARTAG-only isoforms of 4R tau are found in the filament [55].

Under physiological conditions, tau protein stabilizes and maintains the structure of microtubules in neuronal cells by participating in neurite differentiation, growth, and neuronal activity. Under pathological conditions, tau protein undergoes modifications, mainly through hyperphosphorylation, which leads to the formation of toxic aggregates and the loss of tau function with the subsequent destabilization of microtubules [58]. 

Although there is no clear existing genetic evidence linking the dysfunction of tau protein and neurodegeneration, it is well established that neuropathologically, all cases with tau mutations exhibit filamentous tau pathology in the brain. So far, in FTDP17, more than 15 mutations in the tau gene were identified. These mutations are present on either the coding or intronic region located near the splice-donor site of the intron following exon 10. Coding region mutations reduce tau’s ability to interact with microtubules promoting the formation of tau filaments [59]. Several mutations were demonstrated to cause the sulfated glycosaminoglycan-mediated assembly of tau into filaments. Mutations in exon 9 (G272V), exon 12 (V337M), and exon 13 (R406W and G389R) affect all six tau isoforms, while mutations in exon 10 (e.g., P301S, P301L, S305N) affect tau isoforms with four microtubule-binding repeats [60].

It is known that some mutations in the tau gene linked to chromosome 17 (e.g., R406W, V337M, G272V, and P301L) result in increased susceptibility of the tau protein to phosphorylation compared to the wild-type tau. The presence of mutations has been shown to affect protein function in in vitro assays [61,62]. Mutant tau proteins exhibit a reduced ability to bind microtubules and to promote tubulin assembly, with variable effects depending on the mutation [63]. The expression of human tau containing the most common FTDP-17-related mutations (either P301S or P301L) results in motor and cognitive deficits in transgenic mice with the age- and gene dose-dependent development of NFTs [64]. The accumulation of misfolded tau has been proven to be a primary factor in the development of tauopathies; however, the cellular mechanisms involved in tau-related neurodegeneration are still poorly understood [64].

### 3.2. Contribution of the Glial Exosomal Pathway to Tauopathiespo 

A growing body of evidence suggests that protein aggregates involved in tauopathies are capable of a self-sustaining amplification that can spread along neuronal connections, leading to disease progression from specific brain areas in the early phase of the disease to widespread brain regions in the advanced stages. Once the misfolded protein is taken up by the cell, it may act as a template for the misfolding of the endogenous, functional, physiologically folded form of the protein, leading to the production of new aggregates and effectively propagating the disease. This pathway is commonly referred to as “prion-like” spreading [54]. The excessive intracellular formation of tau aggregates leads to cellular damage and death. The release of aggregates from one cell and their uptake by another results in cell-to-cell and region-to-region spreading of the pathology. A recent study revealed that astrocytes promote cell-to-cell spreading of toxic tau aggregates following robust tau pathology in human-induced pluripotent stem cells (iPSCs) and derived neurons and gain neurotoxic features [65]. Although the precise mechanism underpinning this spreading in tauopathies remains to be elucidated, a tauopathy mouse model established that microglia may play a pivotal role in this process by releasing exosomes that can carry pathological misfolded tau, which facilitates the uptake of tau species by neurons. Extracellular tau is also taken up by glial cells [66]. Several studies have attributed the pathological features of glia-derived exosomes to neuropathological conditions such as neurodegenerative diseases, wherein activated astrocytes gain neurotoxic features [58,67]. These changes are often followed by endosomal/lysosomal defects resulting in the increased release and spread of other pathogenic compounds via astrocyte-derived vesicles containing neurodegenerative and neuroinflammatory factors, such as inflammatory complement proteins (C1q, C4b, C3d) [68].

## 4. The Role of Glia in Neurodegenerative Disorders

### 4.1. Astrocyte and Neurodegeneration: An Overview

Several studies have highlighted the importance of astrocyte-dependent non-cell autonomous mechanisms in neurodegenerative disorders [69]. In recent years, fundamental insights into the role of astrocytes in neurodegeneration have been provided. For example, astrocyte activation and the abnormal functioning of these cells have been demonstrated in vitro, in animal models, as well as in the progressive stages of neurodegenerative disorders in humans [3,70,71]. The concept that astrocytes are harmful or protective depending on the physiological or pathophysiological conditions is supported by an increasing number of studies [72,73]. This idea has gained support from a study focusing on amyotrophic lateral sclerosis (ALS), which showed that focal transplantation of glial-restricted precursors (GRPs) exerted a neuroprotective effect along with functional improvement (prolonged survival and attenuated motor neuron loss, as well as slowed declines in forelimb motor) in transgenic rats expressing human mutant superoxide dismutase-1 (SOD^G93A^). 

Of note, the beneficial effects were partially mediated by the major astrocyte glutamate transporter GLT1. Additional putative glia-mediated neuroprotective mechanisms suggested from this study include immune modulation, the suppression of proteoglycan inhibition, and the secretion of neurotrophic growth factors [72,74].

Reactive astrogliosis is a universal response of astrocytes to brain pathologies as diverse as trauma, infection, neurodegeneration, and ischemia that negatively impacts astrocyte function and their properties relevant to neuronal function. Astrogliosis is a spectrum of changes in astrocytes that is manifested as morphological alterations with the increased expression of certain astrocyte markers (GFAP, glial fibrillary acidic protein; vimentin). Remarkable molecular and cellular changes are associated with significant functional impairment of astrocytes, including molecules related to oxidative stress (NOS, nitric oxide synthases; SOD, superoxide dismutase; glutathione), transcriptional regulation (NF-κB, nuclear factor kappa-light-chain-enhancer of activated B cells; STAT3), inflammation (cytokines, growth factors), extracellular matrix and cell–cell interactions (integrins; cadherins; ephrins; metalloproteases), gap juntion proteins (CX43, connexin43), synaptogenesis (TSP, Thrombospondin), vascular regulation (PGE, prostaglandin), or ion and water homeostasis (AQP4, aquaporin-4) [75]. 

### 4.2. Significance of the Astrocyte–Microglia Network in Neurodegeneration and Neuroinflammation

Interest in the involvement of inflammation-induced reactive astrocytes in neurodegenerative diseases has recently grown [76]. Astrocytes in neuropathological conditions can be activated by microglia that secrete several proinflammatory mediators such as nitric oxide and cytokines, which result in neuroinflammation. Type 1 astrocytes (A1; nomenclature analogous to the M1 state of microglial activation) become neurotoxic when activated by microglia and negatively affect neuronal functions by secreting proinflammatory cytokines: IL-1α, TNF-α, and C1q. Recent evidence suggests that the polarization of astrocytes to the A1 phenotype results in a loss of their essential neuroprotective properties. For example, the dysfunctional activation of astrocytes in mouse models of AD impairs neuronal survival through the activation of microglia [67,77]. An in vitro study indicated that retinal ganglion neurons develop significantly fewer synapses when cultured with A1 astrocytes compared with neurons cultured with non-activated astrocytes, which suggests that A1 astrocytes disassemble or fail to maintain synapses. A1 astrocytes release an unidentified toxin causing neuronal death [76]. Furthermore, astrocytes undergo an inflammatory transition after infections and acute injuries, and their response is heterogeneous [78]. In neurodegenerative diseases, their role is complex, as they seem to both alleviate and contribute to pathology. 

## 5. Astrocyte–Neuron Crosstalk in Tau-Dependent Neurodegeneration 

The activation of astrocytes is associated with the onset and progression of virtually all human neurodegenerative diseases [70,79]. In tauopathies, both in patients and animal models, astrogliosis is found in multiple brain regions, including the cerebral cortex and hippocampus. Indeed, astrocyte activation appears to precede neuronal loss, suggesting an important role of astrogliosis in initiation of the disease [76]. Studies using transgenic mouse models expressing human mutant tau exhibit essential features of tauopathies and reveal the crucial role of astrocyte activation in the pathology of the disease. 

Transgenic mice expressing the neuronal Thy1.2 promoter develop neuronal tau phosphorylation and form neurofibrillary tangles (NFTs) in many brain areas, leading to progressive neurodegeneration and behavioral and cognitive dysfunction [5]. Of note, in P301S tau mice, progressive tau aggregation and neuronal loss are associated with ongoing astrogliosis in the superficial layers of the cerebral cortex between two and five months of age [80]. Direct evidence of astrocyte dysfunction was provided by an experiment in which the transplantation of exogenous, neuronal precursor cell-derived astrocytes resulted in a significant reduction in neurodegenerative phenotypes in mice expressing the P301S mutant tau. This indicates that endogenous astrocytes in the tauopathy are deprived of their neuroprotective function and/or may acquire new neurotoxic properties [80]. 

Considerable evidence exists supporting the role of astrocyte-neuron network in tau-dependent neurodegeneration. Studies indicate that astrocytes in mice bearing tau mutation develop pathological changes due to in situ contact with neurons expressing FTD-related mutant tau protein [54]. This deficit is clearly visible after comparison of neuronal survival in vitro either in co-cultures with astrocytes obtained from tau mice (P301SA) or after treatment with astrocyte-conditioned medium (ACM) derived from these astrocytes, with neuroprotective effect of control astrocytes (C57A). In addition, ACM obtained from tau mutant mice astrocytes significantly reduced the expression of presynaptic and postsynaptic markers (synaptophysin and PSD95, respectively) in cortical neuronal cultures, whereas wild-type mouse ACM increased the expression of these proteins. These negative effects on neuronal viability and function have already been found in astrocytes cultured from seven-day-old transgenic animals, an age at which tau aggregation in neurons has not yet been observed in vivo. This indicates that toxic events precede the formation of tau filaments; therefore, the development of astrocyte dysfunction appears to be associated with the earliest manifestations of neuronal tau toxicity. Such a loss of neuroprotective properties was also observed in astrocytes cultured from another transgenic model of tauopathy, P301L mice expressing human 2N4R tau in neurons, indicating that these findings can be generalized as a result of tau pathology [81]. 

Furthermore, the study revealed that astrocytes from the superficial cortical layers of three- and five-month-old P301S mice exhibit several pathological changes. P301S mice expressed higher amounts of GFAP already at three months of age, which persisted until five months of age, while S100β expression was elevated in five-month-old mice. In contrast, there was a reduction in the expression of proteins involved in key neuroprotective functions of astrocytes related to GGC, including GS, GLT1, and GLAST. Moreover, GFAP protein level and astrocytic proliferation were significantly elevated, while and Gln transporters levels were significantly decreased in the lysates from P301SA mice compared to C57A mice. In addition, ACM obtained from tau mutant mice significantly decreased the expression of presynaptic and postsynaptic markers (synaptophysin and PSD95, respectively) and thrombospondin 1 (TSP1), promoting presynaptic and postsynaptic remodeling in cortical neuronal cultures, whereas wild-type mouse ACM increased the expression of these proteins. These results demonstrate that both the in vitro astroglial model and endogenous astrocytes from P301S mice possess an abnormal phenotype starting at an early postnatal age that persists in five-month-old mice, which may explain the loss of neuroprotective function by astrocytes leading to neuronal death in the P301SA model of tauopathy [81] (Figure 2).

A study using several P301S tau mice revealed regional changes in glutamate levels that correlated with the histological assessment of neuropathology, such as pathological tau changes, synapse, and loss of neurons [82]. The decline in the glutamate neurotransmission and mitochondrial dysfunction were also observed in the frontal cortex and hippocampus of aged three × Tg AD mice, developing beta-amyloid plaques and tau aggregates composed of P301L tau [83]. The decreased expression of glutamate-metabolizing enzymes, including glutamate dehydrogenase and glutamine synthetase in astroglia, were noticed in the cerebellum of AD patients [84]. In addition, a more recent study has shown that neuronal activity has a prime role in upregulating gene expression and the function of glutamate transporters in astrocytes [85]. Taken together, it seems that the glutamatergic system is one of the vulnerable points in the reaction between astrocytes and neurons in brain disease and injury, where astrocytes may fail to prevent glutamate excess and neuronal toxicity through the inability to sustain proper glutamate levels.

## 6. Experimental Effects of Mutant Tau P301L: Reduction of Neuronal Survival by Affecting Astrocyte–Neuron Interactions and Disruption of Neuronal-Astrocytic Integrity via the GGC

Our recent study revealed several novel astrocyte-neuron interactions under tau-related pathological conditions [86]. The incubation of primary neuronal cultures with the human recombinant tau and/or control astrocyte-derived medium (ACM) revealed that the presence of ACM in cultures improved the viability of rtau-treated neurons, and significantly higher numbers of neurons were observed after the addition of ACM when cells were incubated with rtau. 

As mentioned in Section 2.2, Gln transport in the CNS is mediated by various sodium-dependent or sodium-independent systems. 

There are carriers that are most likely involved in Gln uptake, while other transporters mediate the bidirectional transfer of this amino acid. It has been revealed that incubation with rtau reduces sodium-dependent Gln uptake by neurons compared to control cultures. Interestingly, the effect of diminished Gln uptake in neurons was reversed when cells were co-incubated with control ACM after exposure to rtau. In contrast, in sodium-independent conditions, the changes in Gln uptake were not reversed, suggesting that most of the astrocyte-derived Gln fuel considered as neuroprotective possesses activity properties of the sodium-dependent amino acid systems. 

As already described, Gln carriers belong to different systems. Under control conditions, Gln is taken up by neurons mainly by system A, and this system appears to be the most sensitive to the addition of rtau to the culture. The addition of ACM-reversed rtau-induced abnormalities in both the total Gln uptake and the system A-mediated uptake. In the case of astrocytes, an increase in the total Gln uptake and a significant increase in the system N-mediated uptake were observed after exposure to 0.1 µg rtau, suggesting a key role for the N system in tau pathological conditions. These observations are consistent with the well-established knowledge that in neurons, Gln is taken up by system A transporters (SNAT1 and SNAT2), while system N mediated-translocation of Gln (mostly via SNAT3 carrier) is highly active in astrocytes [26,48]. 

It is worth noting that the properties of various transporters differ depending on the binding order, stoichiometry, substrate specificity, and other factors. Although many Gln transporters are thought to be expressed constitutively due to their housekeeping role, they are usually regulated or dysregulated by stimuli such as amino acid depletion, cell growth, or, in our case, external pathological stimuli relevant to neurotoxicity and neurodegeneration [32]. Their function may also depend on the cell type and is different in the case of neural cells. In the case of tau-dependent neurodegeneration processes, the study revealed abnormalities in the Gln and Glu transport between neurons and astrocytes, specific to the cortical brain region. Furthermore, the previously described evidence also supports the significant roles of the astrocytic N system and the neuronal A system in tau-mediated pathology relevant to the GGC. 

An important issue in the context of this discussion is why astrocytes respond differently than neurons to the same toxic stimuli and why Gln recycling is affected in tau-dependent neurodegeneration. Tau-related conditions appear to generate a reactive state of astrocytes that drives distinct non-cell-autonomous mechanisms that are either neuroprotective or neurotoxic. One of the changes might be associated with the activation of SNAT3, which belongs to the astroglia-specific N system. The transporter acts in a bidirectional manner, and its supporting role is to recycle Gln and replenish neurons with essential amino acids. Since it is important for neurons to have access to an adequate amount of Gln, SNAT3 supports this need by mediating an efflux of Gln [48]. Such a mechanism may explain the protective effect of ACM observed in our study, which may reflect the SNAT3-dependent regulation of Gln in the culture medium [86]. 

The fact that the GGC is significantly involved in tau-associated pathology can be also confirmed by previous reports. Significant abnormalities in the content of several GGC metabolites, especially the Glu/Gln ratio, were demonstrated in the brain regions of PS19 mice [87]. The disruption of Glu homeostasis in P301L tg mice was noticed in the hippocampus, where the levels of mutant human tau are high, but not in the brain areas with a low expression of transgene [88]. Moreover, a recent study using human iPSC models of FTD patients revealed that glutamate–glutamine homeostasis is disrupted both in neurons and astrocytes [50].

## 7. Conclusions

Astrocytes represent the largest group of the cells within the CNS, and extensive evidence supports their crucial role in neuronal physiology. Astrocytic functional states of astrocytes before injury have the potential to induce neuronal dysfunction. Under pathological conditions, such a tau-dependent neurodegeneration, their function changes, progressing to pathological stages and affecting surrounding neurons. It is well established that in neurodegenerative diseases, reactive astrogliosis occurs parallel to or before the onset of the neuronal dysfunctions and death. The dialogue between neurons and astroglia is dysregulated under tau-dependent pathological conditions, as seen in the disrupted GGC cycle. Abnormalities in the expression of Gln transporters in a mouse model expressing human mutant tau or under conditions of tau pathology in primary cultures of both astrocytes and neurons strongly support the significance of the GGC cycle in tau-dependent neurodegeneration. These pathological events, together with the deprivation of astrocyte neuroprotective properties, seem to be important in regard to gaining a better understanding of the mechanisms of astrocyte–neuron network in tauopathies. 

## Figures and Tables

**Figure 1 ijms-25-03050-f001:**
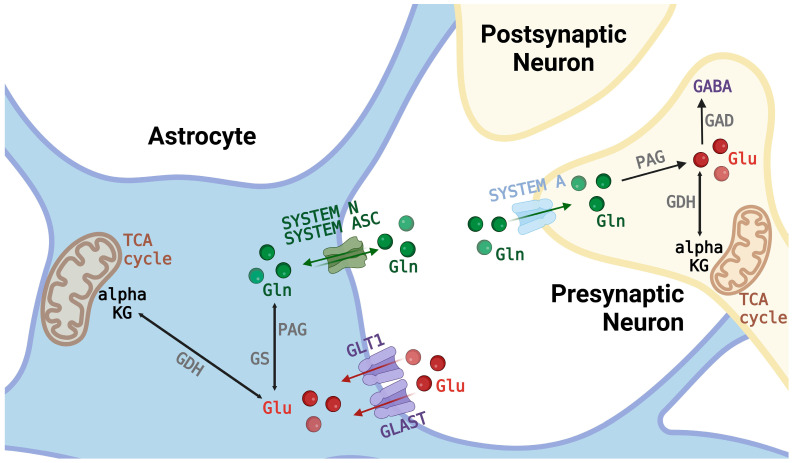
Schematic representation of the Gln and Glu transport and metabolism in astrocytes and neurons. Glu released from synaptic terminals is taken up by surrounding astrocytes via Glu transporters and converted to Gln in GS-mediated reaction. A proportion of Gln is released into the extracellular space by Gln carriers, with a predominant role in the N system. In addition to N system, the release of Gln from astrocytes is mediated by transporters belonging to systems L and ASC. Extracellular Gln is taken up into GABAergic and glutamatergic neurons by the unidirectional system A transporters. Once in neurons, Gln serves as a substrate for the mitochondrial enzyme PAG for the synthesis of Glu that can supply the neurotransmission pool of Glu or can be converted to GABA by GAD or to αKG by GDH.

**Figure 2 ijms-25-03050-f002:**
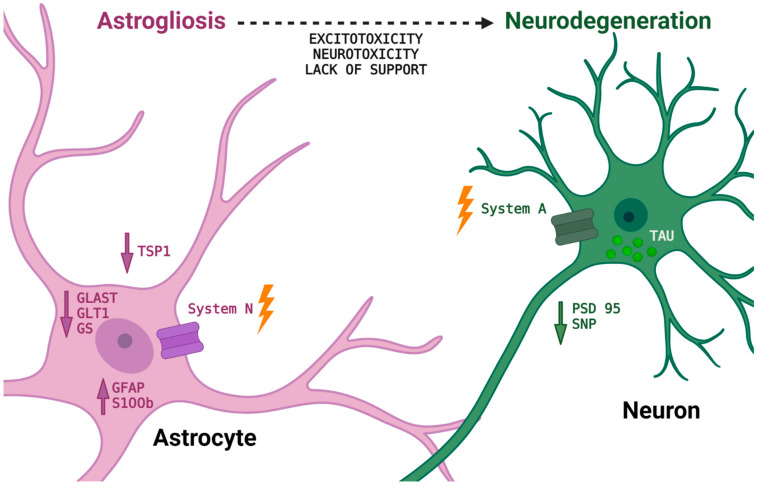
Schematic representation of the disturbed astrocyte–neuron network in tau-dependent neurodegeneration based on the previously described results [81]. Astroglia after exposure to the mutant tau-expressing neurons, acquire functional deficit resulting in loss of neurosupportive properties and/or gain of neurotoxic function. One of the pathological events is associated with overexpression of GFAP and S100β protein as a consequence of the astrocytic phenotype reactivity. Moreover, astrocytes became deprived of key molecules that support glutamate (GLAST, GLT-1) and glutamine homeostasis (System N, System A carries) and synaptogenesis relevant processes (TSP-1, PSD95, SNP). Up arrows indicate targets that are increased; down arrows represent declined targets; lightning arrows mean deregulation of the glutamine transporting systems function in the astrocytic and neuronal tau-mediated neurodegeneration. For more details, see Section 5 and Section 6.

**Table 1 ijms-25-03050-t001:** Glutamine transporters operating in the astrocytes and neurons functionally involved in glutamate-glutamine cycle (GGC), their cellular specificity, and their activity; approach to study glutamine transport using competitive amino acids substrates.

Amino Acids Transporting System in the CNS	Variant(s); Cellular Specificity; Noticeable Activity	Competitive Substrate Used for the Functional Study	References
N	SNAT3, SNAT5; astrocyte	MeAIB, Thr, Leu	[32,33]
A	SNAT1, SNAT2; astrocyte/neuron	Thr, Leu, His	[34]
ASC	ASCT1; astrocyte	MeAIB, His, Leu	[32]
L	LAT1, LAT2; neuron	MeAIB, Thr, His	[35]

## Data Availability

Not applicable.

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
