# Peer review of "Astrocyte–Neuron Interaction via the Glutamate–Glutamine Cycle and Its Dysfunction in Tau-Dependent Neurodegeneration"

_ijms, 2024, doi:10.3390/ijms25053050_

Round 1

Reviewer 1 Report

Comments and Suggestions for Authors

This review presents a summary of current literatures related to the role of astrocytes in tau-mediated neurodegenerative diseases. Specifically, the authors focused on the classical glutamate-glutamine cycle between astrocytes and neurons. This is a relevant, important and interesting topic that is most often neglected in view of many diverse types of neuroglia interactions. 

While the report is comprehensive, logical and organized, the emphasis on the link between glutamine/glutamate/GABA level and tauopathy could be strengthened. Some suggests to consider:

- discuss whether astroglial dysfunction, besides generally preceding neurodegeneration, might be considered the mediator of disease progression or a first line response.

- going more downstream, discuss likely benefits of maintaining the glutamate glutamine cycle in the neurodegenerative disease context. Eg. Is it purely metabolic or might it be to support synaptic activities?

- the use of glutamine supplements to alleviate symptoms of neurodegenerative diseases could be mentioned.

Author Response

We agree with reviewers that we missed some important informations. In our current version we are describing suggested issues (Section 4.2, 5).

Reviewer 2 Report

Comments and Suggestions for Authors

Reviewer comments

The present review article discusses the role of abnormalities in the expression of the glutamate-glutamine transporter cycle, which inhibits the interaction of astrocytes and neuron networks and leads to tau-dependent neurodegeneration.

The review topic is very interesting and has the potential to be published at the current time however MS has many shortcomings. The MS has very basic information (too basic information in a very simple way that is not needed) and does not provide potent research-based data to show the molecular mechanisms that link Astrocyte-neuron interaction via the glutamate-glutamine cycle and its dysfunction in tau-dependent neurodegeneration. MS needs to be revised thoroughly.

Scientific comments

1.      Most of the references are from before 2020. Replace the very old references with new ones.

2.      In Figure 1, mention the full form of GDH.

3.      Discuss lines 90-91 briefly with a suitable reference.

4.      Are the terms L system and L transporter the same? If yes, use any one pattern throughout the revised MS.

5.      Provide a suitable reference for lines 175–176.

6.      Line 184, the word Parkinson's disease (PD) has been used only a single time in MS so there is no need for an abbreviation.

7.      Rewrite the sentence for lines 200–201.

8.      Provide a suitable reference for lines 211-212.

9.      Section 3.1. is very basic and lengthy. Rewrite briefly.

10.  Summarize the research-based data of four mutations mentioned in lines 209–210 (R406W, V337M, G272V, 209, and P301L) with their mutant role in AD/tauopathy.

11.  Discuss the following papers:

https://doi.org/10.1016/j.redox.2023.102690

https://doi.org/10.3389/fncel.2022.1037641

https://doi.org/10.1186/s13041-020-00658-6

Typo error

1.      Delete the full stop from the title.

2.      Delete full stop from subtitles 3.2 and 4.2.

3.      Line 127, add full stop after amino acids.

4. Check the pattern of writing tau or Tau. use the same pattern throughout the MS.

Author Response

  1. Most of the references are from before 2020. Replace the very old references with new ones.

In current version we are providing new references and we keep proper balance between old and new references ratio. Due to the subject of this review, large number of references providing original data/discovery.

  1. In Figure 1, mention the full form of GDH.

We corrected this issue.

  1. Discuss lines 90-91 briefly with a suitable reference.

We corrected this issue according to the reviewer suggestion.

  1. Are the terms L system and L transporter the same? If yes, use any one pattern throughout the revised MS.

This terms are different.

  1. Provide a suitable reference for lines 175–176.

It was corrected.

  1. Line 184, the word Parkinson's disease (PD) has been used only a single time in MS so there is no need for an abbreviation. It was corrected.

  1. Rewrite the sentence for lines 200–201.

It was corrected.

  1. Provide a suitable reference for lines 211-212. It was corrected.

  1. Section 3.1. is very basic and lengthy. Rewrite briefly.
  2. Summarize the research-based data of four mutations mentioned in lines 209–210 (R406W, V337M, G272V, 209, and P301L) with their mutant role in AD/tauopathy.

9/10 we rewrite section 3.1. according to the reviewer suggestions.

  1. Discuss the following papers:

https://doi.org/10.1016/j.redox.2023.102690 (section 1.1

https://doi.org/10.3389/fncel.2022.1037641 (section 1.1

https://doi.org/10.1186/s13041-020-00658-6 (section 2.2.

These papers are included to the current version with a brief descriptions.

Typo error

  1. Delete the full stop from the title.)
  2. Delete full stop from subtitles 3.2 and 4.2.
  3. Line 127, add full stop after amino acids.
  4. Check the pattern of writing tau or Tau. use the same pattern throughout the MS.

We corrected typo errors.

Round 2

Reviewer 2 Report

Comments and Suggestions for Authors

The authors have justified all the previous comments and discussed the suggested references. 

paper can be accepted in its present form.